# Causal Path Tracing in Transformers

## Abstract

We propose a causal path tracing framework to understand how information causally flows through the internal structures of transformers for a given decision. By unfolding each block into a causal graph of path nodes and applying a *minimality-based subset search*, our method identifies all possible causal paths within each block, with polynomial-time complexity on average. Furthermore, we demonstrate the reliability of a *union-based causal path reference strategy*, enabling efficient and reliable causal tracing throughout the model. The key contributions of this work are: (1) an automated, efficient framework for causal path tracing that exhaustively searches paths along direct dependencies; (2) theoretical and empirical validation demonstrating exhaustive search with polynomial-time complexity on average; (3) experimental findings showing that self-repair effects occur far less frequently along the identified causal paths, that certain paths are uniquely activated for specific classes, and that the traced paths are both accurate and faithful.

## 1 Introduction

With the success of transformers [1] across language and vision, interest has grown in understanding their internal mechanisms beyond their black-box nature, especially to enable safer deployment in high-stakes applications such as healthcare, law, and education. Mechanistic interpretability aims to identify specific components within the model, such as attention heads or MLPs, that contribute to its behavior. Building with mathematically grounded circuit discovery in simplified settings [2], recent efforts have incorporated Pearl's causal theory [3], employing ablation-based interventions to trace which parts of the network support particular outputs.

Depending on the granularity of analysis, prior work can be classified into: node-level patching [4, 5, 6, 7, 8, 9], which identifies the role of individual input features; edge-level patching [10, 11, 12], which examines the influence of neighboring feature pairs with direct computational dependencies; and path-level patching [13, 14, 15], which investigates the contribution of distant feature pairs connected through multiple accumulated dependencies.

Recent work [16] has shown that ablation-based methods often fail to estimate true causal effects due to self-repair (or backup behavior), where later components compensate for earlier ablations. This implies that, when unablated components lie between the target and the decision, internal explanations may be misattributed. To address this, one solution is to iteratively evaluate each component conditioned on priorly identified causal components along direct computational dependencies; here, we refer to as *causal referencing*. However, prior node- or edge-level approaches cannot fully support causal referencing over all relevant combinations; though sequentially feasible, it remains inaccurate. In contrast, path-level approaches can in principle support this, but due to their combinatorial complexity, existing studies typically rely on hypothetical tests [13] or assess a single subpath manually [14, 15], making it infeasible to capture a full explanation for a decision in time. (Table 1)

To address this obstacle, we propose an automated and efficient framework for tracing causal paths given a decision. Specifically, we begin by unfolding all possible paths within each block of a

| Approach | Patching | Path Tracing for Decision | |
|---|---|---|---|
| | | **Feasibility** | **Reliability** |
| [4, 5, 6, 7, 8, 9] | Node | ✓ backward chaining only | ✗ no causal referencing |
| [10, 11, 12] | Edge | ✓ backward chaining only | ✗ no causal referencing |
| [13] | Path | ✗ hypothetical only | ✓ full coverage |
| [14, 15] | Path | ✗ manual subpath | ✓ full coverage |
| **Ours** | Path | ✓ polynomial on average | ✓ full coverage |

Table 1: **Comparison of patching methods for path tracing in a given decision.** Feasibility refers to empirical applicability for a given decision; reliability to its theoretical guarantees.

transformer, interpreted as a causal graph, into path nodes. Then, by introducing a minimality-based subset search strategy for identifying all possible causal path node combinations per block, we reduce the inherently exponential complexity to polynomial time on average. Furthermore, to enable efficient block-wise tracing, we demonstrate that referencing the union of causal paths identified in preceding blocks not only makes this feasible but also ensures reliability.

Our approach reveals that self-repair occurs primarily outside the identified causal path; thus, the path contains information essential to the decision and not easily replaced, reflecting its critical role. Moreover, we found that there exist causal paths uniquely associated with specific classes. These paths are activated only for their corresponding classes, serving class-specific roles within the model. Taken together, our results show that the proposed method faithfully and accurately explains model behavior under empirical evaluation.

## 2 Methodology

### 2.1 Preliminaries

To proceed, we introduce the definitions used throughout this work. Our goal is to reveal and explain internal components for decision by efficiently tracing causal paths. To enable this, following Pearl's causal theory [3], we interpret the transformer as a causal graph, as formalized in Definition 1. Based on this interpretation, we define the causal path for a given model decision through Definitions 2 to 4, where Definition 4 is adapted from the Halpern–Pearl definition of actual causality [17, 18].

**Definition 1** (Transformer as Causal Graph). *We say that a transformer is a **causal graph** $\mathcal{G} = (\mathcal{V}, \mathcal{E})$, where each node $v \in \mathcal{V}$ denotes an internal component (e.g., intermediate feature) and each edge $v_i \rightarrow v_j \in \mathcal{E}$ indicates a direct computational dependency, if it satisfies the following conditions:*

*(1.a) **Directed Acyclic Graph:** Its internal computation proceeds layer by layer in a forward direction without cycles, which naturally forms a directed acyclic graph structure.*

*(1.b) **Markov:** Each node is deterministically computed from its parent nodes. This ensures that each node is conditionally independent of its non-descendants, given its parents.*

*(1.c) **Causal Sufficiency:** All nodes involved in its internal computation are observable, with no latent confounders or hidden common causes among nodes.*

**Definition 2** (Model Decision). *Let $y \in \mathbb{R}^{\mathcal{C}}$ be the model's output over $\mathcal{C}$ classes. We define the **model decision** as the index $c^*$ such that $y^{(c^*)} > y^{(i)}$ for all $i \neq c^*$, i.e., the strict argmax.*

**Definition 3** (Causal Path). *Given a transformer interpreted as a causal graph $\mathcal{G}$, we define a **causal path** as a sequence of causal node sets $\mathcal{P} = (V_1, V_2, \dots)$, where each $V_i \subseteq \mathcal{V}$ is a **causal node set**, and every node $v \in V_i$ is connected via a directed edge to either the model input, the model output $y$, or a node in another causal node set $V_j$ with $j \neq i$.*

**Definition 4** (Causal Node Set). *Given a transformer interpreted as a causal graph $\mathcal{G}$, a causal subpath reference $P \subseteq \mathcal{P}$ connecting a node set $V \subseteq \mathcal{V}$ to the output, and an off-path node set $\hat{V}$ such that $P \cup \hat{V}$ equals the set of all nodes between $V$ and the output, and $P \cap \hat{V} = \emptyset$, we say that $V$ is **causal** for the decision if the following conditions are satisfied:*

*(4.a) **Necessity (Counterfactual):** Let $V'$ denote that $V$ is intervened on to take a different value, and let $\hat{V}'$ be defined analogously for $\hat{V}$ to causally isolate $V$. Under these interventions, the output $y'$ satisfies $\arg\max_i y'^{(i)} \neq c^*$, where $c^*$ denotes the decision without any intervention.*

*(4.b) **Sufficiency (Contingency):** Given $V$, even if nodes in $P$ are perturbed due to an intervention resulting in $\hat{V}'$, the output $y'$ satisfies $\arg\max_i y'^{(i)} = c^*$.*

*(4.c) **Causal Minimality:** $V$ is minimal; no strict subset of $V$ satisfies both (4.a) and (4.b).*

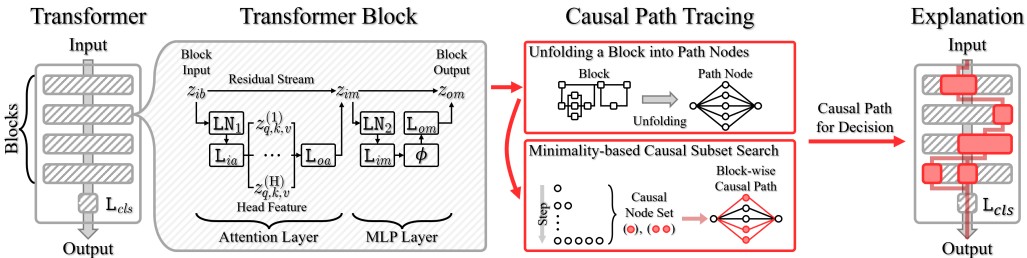

Figure 1: **Overview of our causal path tracing.** L: linear layer, LN: layer normalization, $z$: feature.

Having established the definitions with respect to causal paths, we present the structural Property 1 that highlights their recursive nature: under a given decision, any causal node set must have at least one parent node set that is also causal. This recursive property enables us to identify exhaustive causal paths by ensuring that causal influence can be traced backward through successive parent node sets.

**Property 1** (Causal Edge: Existence of Causal Parent for Any Causal Node). *In a transformer interpreted as a causal graph $\mathcal{G}$, each child node has a direct computational dependency (i.e., edges) with its parent nodes. Thus, under sufficient intervention, for every child node in $\mathcal{G}$ that belongs to a causal node set (possibly not minimal), at least one of its parent node sets is also a causal node set (possibly not minimal), given the decision. Here, "possibly not minimal" indicates that the node set may satisfy only the intervention-based conditions (Conditions (4.a) and (4.b)) without the minimality condition (Condition (4.c)). Although such a set may not strictly qualify as a causal node set under the full definition, the minimality condition is dependent on the other two and is not required for this property to hold.*

**Example 1.** *Consider a transformer interpreted as $\mathcal{G}$, containing only a node $v_c$ and the set $V_p$ of all parent nodes of $v_c$. Suppose that $v_c$ belongs to a causal node set (i.e., the causal node set consists of $v_c$ alone). Since $v_c$ is deterministically computed from $V_p$, intervening on all of $V_p$ leads to a different decision, satisfying Condition (4.a). Moreover, if $V_p$ is unchanged, the decision remains the same since there is no off-path node set (i.e., $\hat{V} = \emptyset$), thereby satisfying Condition (4.b). Thus, $V_p$ satisfies both conditions in Definition 4, implying that it is, at least, a causal node set.*

## 2.2 Intervention for Causal Isolation

As established in the preceding definitions, applying an intervention to isolate a node set is essential for identifying its causal influence in transformers. While there are various possible forms of intervention, it is not always clear whether they guarantee causal isolation under our setting. To address this, we formally define a *sufficient intervention* in Definition 5, to serve as a basis for assessing whether an intervention achieves reliable causal isolation in our framework.

**Definition 5** (Sufficient Intervention). *Given a transformer interpreted as a causal graph $\mathcal{G}$, we say that a node set $V$ is **sufficiently intervened** to $V'$ if the following conditions are satisfied:*

*(5.a) **Causal Structural Isomorphism:** The graphical structure of $V$ in $\mathcal{G}$, namely the adjacency structure between $V$ and its neighboring nodes, must differ from that of $V'$, and their corresponding mathematical structures (i.e., structural equations) must likewise differ. This reflects a one-to-one correspondence between graphical and mathematical structures.*

*(5.b) **Causal Edge Validity:** Given that Property 1 holds in $\mathcal{G}$, it must still hold even after an intervention on $V$. That is, the intervention must not violate the conditions specified in Definitions 1 to 4 with respect to causal paths in $\mathcal{G}$.*

*(5.c) **Intervention Controllability:** An intervention must not be parametrically non-controllable; its effect must remain interpretable, rather than being overwhelmed by the parameters of the intervention, such as stochastic variations within them.*

Among possible intervention strategies within transformers, a straightforward approach is to add noise directly to the target node (DIRECT NOISE). However, such perturbation fails to satisfy Condition (5.a). Another commonly used strategy, as in prior works such as [6], involves forwarding a noise-perturbed token embedding through the model to obtain a corrupted version of the target node (NOISE TOKEN);

123 however, this violates Condition (5.c). A naive alternative, such as zero-masking (ZERO MASK), also
124 proves inadequate, as it breaks Condition (5.b) by distorting Property 1.

125 Since these strategies fail to satisfy the required conditions in our setting, we instead adopt a method
126 that intervenes on the target node by resampling from alternative token embeddings (TOKEN RESAM-
127 PLING). This approach satisfies all three conditions for a sufficient intervention, as demonstrated by
128 the example in Appendix.

## 2.3 Unfolding Transformer Block

130 In this section, we introduce a mathematical formulation of paths within a standard transformer. We
131 begin by representing paths in a single block to establish the idea of our approach. Subsequently, we
132 extend this formulation to cover all possible paths from the given input to the output.

133 Given an input $x$, our goal is to identify *which structures* within the transformer contribute to the
134 decision as causal paths. This requires identifying the causal node sets from the decision, which in
135 turn involves exploring the model in the backward direction. To this end, we first decompose a single
136 block into circuits, i.e., paths, as follows (notation is provided in Figure 1):

$$\left[ [z_q^{(h)}]_{h=1}^H; \ [z_k^{(h)}]_{h=1}^H; \ [z_v^{(h)}]_{h=1}^H \right] = \mathsf{L}_{ia}(\mathsf{LN}_1(z_{ib})),$$
$$z_{oa} = \mathsf{L}_{oa}([\,\mathrm{softmax}(z_q^{(h)} z_k^{(h)\top}/\sqrt{d_h}) z_v^{(h)}\,]_{h=1}^H),$$
$$z_{im} = z_{ib} + z_{oa},$$
$$z_{om} = \mathsf{L}_{om}\left(\phi(\mathsf{L}_{im}(\mathsf{LN}_2(z_{im})))\right),$$
$$z_{ob} = z_{im} + z_{om}, \tag{1}$$

137 where $z_{ib}, z_{oa}, z_{im}, z_{om}, z_{ob} \in \mathbb{R}^{T \times d_m}$ and $z_q^{(h)}, z_k^{(h)}, z_v^{(h)} \in \mathbb{R}^{T \times d_h}$, with $T$ denoting the number
138 of tokens, $d_m$ the model dimension, and $d_h = {}^{d_m}/H$. Here, to identify the structures causally involved
139 in the decision from the input, we treat the block input $z_{ib}$ as a single node. If the block output $z_{ob}$
140 can be unfolded with respect to $z_{ib}$, this allows us to capture the structures in a path-wise manner
141 $(z_{ib} \sim z_{ob})$, all at once, rather than laboriously analyzing them one by one in a structure-wise fashion.

142 However, due to the presence of non-linear functions, i.e., softmax and GeLU $\phi$, it is nontrivial
143 to decompose the above equations into a single unified expression. To address this, we employ a
144 minor computational trick that rewrites the non-linear functions in the form of Hadamard products:
145 $\mathrm{softmax}(z/\sqrt{d_h}) = z \odot D_\alpha$ and $\phi(z) = z \odot D_\beta$, where the scaling factors $D_\alpha$ and $D_\beta$ are treated
146 as fixed values once computed from the input $z$. In addition, we apply a similar simplification to layer
147 normalization by treating its input-dependent statistics, mean and variance, as fixed after computation:
148 $\mathsf{LN}(z) = z W_{ln}^\top + b_{ln}$. Together, these interpretations allow us to express the block in a form that
149 structurally resembles a composition of linear operations and element-wise products, as follows:

$$z_{oa} = (\sum_{h=1}^H z_q^{(h)} z_k^{(h)\top} \odot D_\alpha z_v^{(h)} W_{oa}^\top) + b_{oa},$$
$$z_{om} = z_{oa} W_{ln_2}^\top W_{im}^\top \odot D_\beta W_{om}^\top + z_{ib} W_{ln_2}^\top W_{im}^\top D_\beta W_{om}^\top$$
$$\qquad + b_{ln_2} W_{im}^\top \odot D_\beta W_{om}^\top + b_{im} \odot D_\beta W_{om}^\top + b_{om},$$
$$z_{ob} = z_{ib} + z_{oa} + z_{om}$$
$$= \underbrace{z_{ib}}_{\text{Residual Only (\textit{1 Path})}} + \underbrace{\sum_{h=1}^H z_q^{(h)} z_k^{(h)\top} \odot D_\alpha z_v^{(h)} W_{oa}^\top + b_{\text{attn}}}_{\text{Attention Only (\textit{H Paths})}} + \underbrace{z_{ib} W_{ln_2}^\top W_{im}^\top D_\beta W_{om}^\top + b_{\text{mlp}}}_{\text{MLP Only (\textit{1 Path})}}$$
$$+ \underbrace{\sum_{h=1}^H z_q^{(h)} z_k^{(h)\top} \odot D_\alpha z_v^{(h)} W_{oa}^\top W_{ln_2}^\top W_{im}^\top \odot D_\beta W_{om}^\top + b_{\text{attn+mlp}}}_{\text{Attention+MLP (\textit{H Paths})}}, \tag{2}$$

150 Here, $b_{\text{attn}}, b_{\text{mlp}}$, and $b_{\text{attn+mlp}}$ represent the terms in the block output $z_{ob}$ that do not directly involve
151 the block input $z_{ib}$ (the full derivation is provided in Appendix). By unfolding $z_{ob}$ with respect to $z_{ib}$,
152 we obtain a set of additive terms, which can be grouped into distinct paths depending on whether they
153 contain $z_{ib}$ as a multiplicative factor. Ultimately, we can *treat each of these paths as a single node* to
154 assess its causal contribution.

---
**Algorithm 1** Minimality-based Causal Subset Search per Block
---
1: **Input:** A path node set $V_p = [v_1, \ldots, v_n]$ from a specific block (i.e., additive terms within the block), a subgraph $\mathcal{G}_c$ (downstream blocks of $V_p$ in the transformer $\mathcal{G}$), causal subpaths $P$ connecting $V_p$ to the decision $c^*$ in the output $y$, and an off-path node set $\hat{V}$ such that $P \cup \hat{V}$ equals all nodes in $\mathcal{G}_c$ and $P \cap \hat{V} = \emptyset$

2: **Output:** $V_{\text{out}} = \{V_1, V_2, \ldots\}$, where each $V_i \subseteq V_p$ satisfies Conditions (4.a), (4.b), and (4.c)

3: $V_{\text{out}} \leftarrow \emptyset$

4: **for** $s = 1$ to $n$ **do**                                                  ▷ Subset size

5:     **for** each $V \subseteq V_p$ such that $|V| = s$ **do**

6:         **if** $V_i \subseteq V$ for some $V_i \in V_{\text{out}}$ **then**

7:             **continue**                      ▷ Fail Condition (4.c) (causal minimality)

8:         **end if**

9:         Intervene on $V \rightarrow V'$, and on $\hat{V} \rightarrow \hat{V}'$

10:         Let $y' \leftarrow$ model output under $(V', \hat{V}', P)$        ▷ for Condition (4.a) (necessity)

11:         Let $y'' \leftarrow$ model output under $(V, \hat{V}', P)$        ▷ for Condition (4.b) (sufficiency)

12:         **if** $\arg\max_i y'^{(i)} = c^*$ **or** $\arg\max_i y''^{(i)} \neq c^*$ **then**

13:             **continue**                          ▷ Fail Condition (4.a) or (4.b)

14:         **end if**

15:         $V_{\text{out}} \leftarrow V_{\text{out}} \cup \{V\}$             ▷ Satisfies Conditions (4.a), (4.b), (4.c)

16:     **end for**

17: **end for**

18: **return** $V_{\text{out}}$
---

Note that we omit the unfolding of $z_q^{(h)}$, $z_k^{(h)}$, and $z_v^{(h)}$ for brevity, as they are linear functions of $z_{ib}$ via $W_{ln_1}, W_{ia}$ and follow the same path structure. Furthermore, we assume that bias terms explicitly excluding $z_{ib}$ propagate uniformly their influence across all paths through their originating layers.

## 2.4 Minimality-based Causal Subset Search per Block

As shown earlier, all paths within a block can be decomposed into additive terms, each treated as an individual node. Based on this, we perform a block-wise backward search for causal node sets to trace the causal path for a given decision. Here, since path-level interactions must be considered, all possible combinations of path nodes within each block need to be evaluated. However, a brute-force approach incurs a complexity of $O(2^n)$, as this subset search problem is NP-complete, making it impractical for large-scale search. Although NP-complete problems cannot be solved in polynomial time in the worst case, we propose a strategy based on Condition (4.c) that enables polynomial-time search on average.

The core idea, based on Condition (4.c), is that a causal node set must be minimal. That is, if a subset $V \subseteq V_p$ is identified as a causal node set, where $V_p$ denotes the set of all path nodes within a block, then any superset of $V$ cannot be minimal and thus does not need to be evaluated. Building on this, our search strategy proceeds in steps by subset size, starting from the smallest. As illustrated in Algorithm 1, causal node sets identified at smaller steps are used to prune the search space at larger steps by eliminating supersets that violate minimality. This strategy leads to an average-case time complexity that is polynomial in practice, as formally analyzed in Theorem 1 (proof in Appendix).

**Theorem 1** (Expected Time Complexity of Minimality-based Subset Search). *Consider a minimality-based subset search over $n$ nodes, where each subset is independently selected as a causal node set with probability $p$. Then, the expected number of subset evaluations over all subsets is bounded by:*

$$n + (1-p) \times \sum_{s=2}^{n} \max\left(0, \binom{n}{s} + \sum_{i=1}^{s-1} \sum_{m=1}^{\lfloor p\binom{n}{i} \rfloor} (-1)^m \binom{p\binom{n}{i}}{m} \binom{n-mi}{s-mi}\right). \quad (3)$$

*Given this, the expected time complexity grows approximately as:*

$$O\left(n^{\lfloor \log_2\left(\frac{1}{p}+2\right) \rfloor}\right). \quad (4)$$

**Remark 1.** *Although the exact value of $p$ is unknown, the time complexity, depending on $p$, is polynomial in the best and average cases. For example, when $p = 1$, all subsets of size $s \geq 2$ are pruned, so only singleton subsets are evaluated, resulting in a time complexity of $O(n)$. However,*

---

**Algorithm 2** Unfolded Block-wise Causal Path Tracing

---

1: **Input:** A transformer $\mathcal{G}$ with $D$ blocks, and a model output $y$ with decision $c^*$ for a given input
2: **Output:** Causal paths $\mathcal{P} = (V_{\text{out}}^{(D)}, \ldots, V_{\text{out}}^{(1)})$, a sequence of causal node sets identified per block
3: $P \leftarrow \{\mathrm{L}_{cls}\}$       $\triangleright$ By Property 1, the classifier $\mathrm{L}_{cls}$ serves as the initial causal path reference
4: $\mathcal{P} \leftarrow \emptyset; \quad \mathcal{G}_c \leftarrow \{\mathrm{L}_{cls}\}$
5: **for** $j = D$ to 1 **do**                  $\triangleright$ Iterate backward through transformer blocks
6:      Let $V_p^{(j)} \leftarrow$ unfolded path nodes in block $j$
7:      $V_{\text{out}}^{(j)} \leftarrow \text{MIN\_SEARCH}(V_p^{(j)}, \mathcal{G}_c, P, c^*)$            $\triangleright$ See Algorithm 1
8:      $P \leftarrow \bigcup V_{\text{out}}^{(j)}$             $\triangleright$ Update causal path reference (see Theorem 2)
9:      $\mathcal{P} \leftarrow \mathcal{P} \cup V_{\text{out}}^{(j)}; \quad \mathcal{G}_c \leftarrow$ block $j$
10: **end for**
11: **return** $\mathcal{P}$

---

since the problem is fundamentally NP-complete, exponential complexity is unavoidable in the worst case. Nonetheless, such worst-case scenarios occur only infrequently; for example, when $p \leq \frac{1}{2^n - 2}$, causal node sets are rarely selected at each step, requiring exhaustive search over all subset combinations and leading to a time complexity of $O(2^n)$.

## 2.5 Unfolded Block-wise Causal Path Tracing

In this section, we extend the minimality-based causal subset search from a single block to the entire transformer. We traverse blocks backward, identifying causal node sets and updating the causal path reference $P$ at each step. Using each causal set individually as $P$ is computationally expensive, as it requires repeated searches. Instead, we use their union as the reference, which significantly reduces the cost. As in Theorem 2, the union-based strategy ensures that reliability converges to 1 (proof in Appendix), indicating near-complete causal coverage. Algorithm 2 outlines the full procedure.

**Theorem 2** (Causal Union Reference Reliability). *Consider a minimality-based subset search over $n$ nodes, where each subset is independently selected as a causal node set with probability $p$. Suppose that a collection of such sets, $V_{out}^{(j+1)} = \{V_i^{(j+1)}\}_{i=1}^{k}$, is identified from the $(j+1)$-th block, i.e., the one directly downstream. Their union, denoted as $P = \bigcup_{i=1}^{k} V_i^{(j+1)}$, serves as the causal subpath reference for the minimality-based subset search in the $j$-th block. Let $s_{avg}$ denote the average size of the $k$ causal node sets in $V_{out}^{(j+1)}$. Then, the reliability of the resulting causal node set obtained using $P$ is given by:*

$$p + (1-p)\left(1 - \left(1 - \frac{s_{avg}}{n}\right)^k\right)^n \to 1 \tag{5}$$

# 3 Experiments

## 3.1 Models, Datasets, and Baselines

We conduct experiments on five transformer models: three language models (GPT2-xs [19], Pythia-14m and Pythia-1b [20]) and two vision models (ViT-tiny [21] and DeiT-tiny [22]). For language tasks, we use the KNOWNS1000 [6] and T-REx [23, 24] datasets. For vision tasks, we evaluate on IMAGENET [25] and OFFICEHOME [26]. Due to space constraints, further results are provided in Appendix.

As summarized in Table 1, we compare against existing methods that are feasible for decision-level path tracing. To enable fair comparison with our method, all baselines are extended under a backward chaining framework, assuming that residual connections are always present—even when reliability conditions are not met.

Specifically, $\text{NT}_1$ and $\text{NT}_{10\%}$ are adaptations of the node-level patching method from [6], referred to as Node-level patching-based Tracing (NT), where the top-1 node ($\text{NT}_1$) or the top 10% of nodes ($\text{NT}_{10\%}$), ranked by their estimated effect within each block, are selected as decision paths. $\text{ET}_{\text{all}}$ and $\text{ET}_{\text{cls}}$ are based on the edge-level patching method from [12], referred to as Edge-level

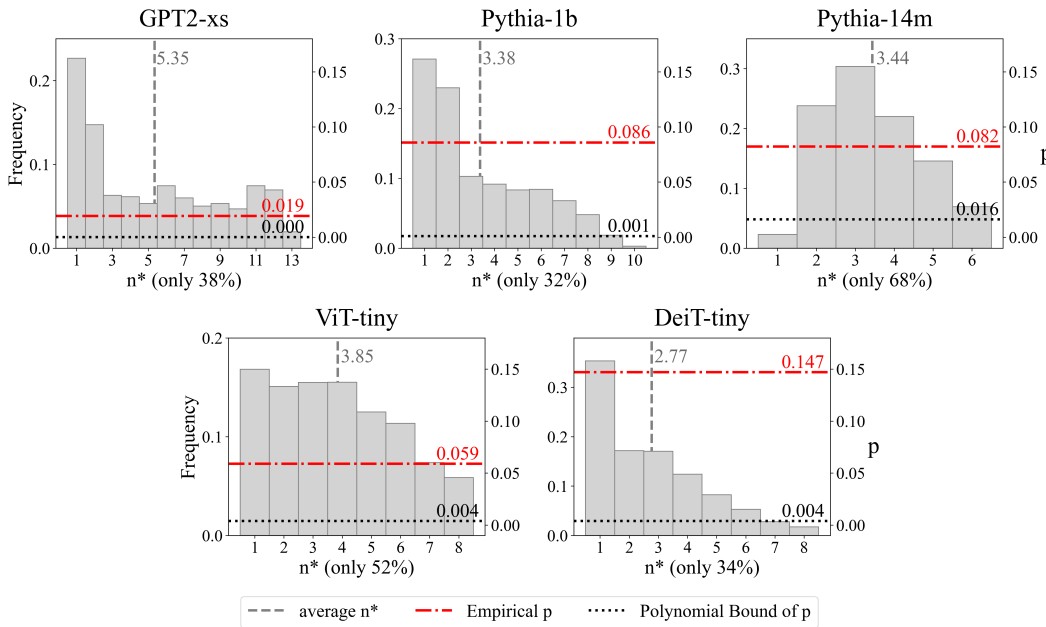

Figure 2: **Comparison of empirical time complexity.** Causal path tracing under our method runs in polynomial time across models. Each subplot shows the reduced search space (in parentheses); $n^*$ is the maximum step reached by the minimality-based search (i.e., the largest $s$ such that the term in Equation (3) is nonzero); the empirical $p$ estimated from the average $n^*$ (see Theorem 1); and the polynomial bound of $p$, which is the theoretical lower bound required to ensure polynomial-time search (see Remark 1). Language models use T-Rex; vision models use ImageNet.

patching-based Tracing (ET), which assumes task-level edge attribution. Here, a "task" is defined as either the entire dataset ($ET_{all}$) or a single class ($ET_{cls}$).

Note that path-level patching methods are not included in the comparison, as no existing method feasibly enumerates all decision paths for a given output—our method is the first to make this feasible. We refer to our approach as Causal Path Tracing (CPT). Implementation details are in Appendix.

## 3.2 Results

**Minimality-based search converges empirically in polynomial time; furthermore, it reveals how models rely on path-level reasoning** Figure 2 presents the empirical time complexity of our causal path tracing procedure across models. Each subplot shows the distribution of $n^*$, defined as the final step in the minimality-based search where no further superset remains due to pruning by already selected causal node sets. The average $n^*$ is used to estimate the empirical probability $p$ that a randomly selected subset is causal (see Theorem 1), which is then compared against the theoretical lower bound required for polynomial-time search (see Remark 1).

In all models, the empirical $p$ exceeds the theoretical threshold, confirming that the proposed search converges in polynomial time in practice, as predicted. Notably, the search typically completes in few steps, with pruning often concluding well before the midpoint of the search space.

The distribution of $n^*$ also reveals how the model leverages internal structure for decision making: a small $n^*$, especially when concentrated near one, indicates that the model relies primarily on the strength of individual paths; in contrast, a larger or more dispersed $n^*$ suggests that reasoning involves interactions among multiple paths rather than relying on any single strong one.

**Causal path components exhibit lower self-repair, suggesting irreplaceable decision signals** We compare self-repair scores between attention heads on the causal path and those off the path, as identified by our tracing method. Following the prior work [16], we categorize components based on whether they belong to the traced causal path and measure their self-repair accordingly.

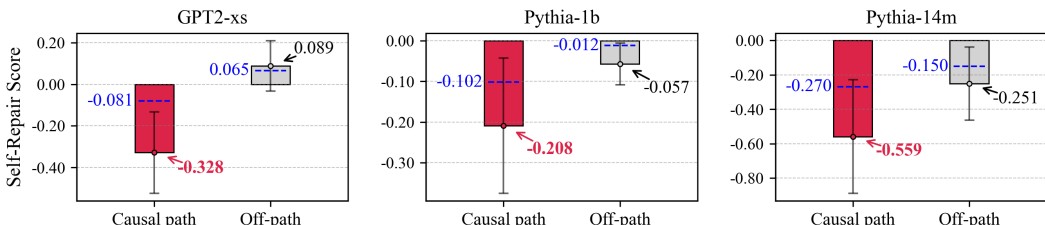

Figure 3: **Self-repair scores on causal path vs. off-path components.** Each bar shows the mean (dot with arrow) and standard deviation (error bar); medians are shown as blue dashed lines. Lower scores indicate less self-repair. Results are averaged over KNOWNS1000 and T-REX.

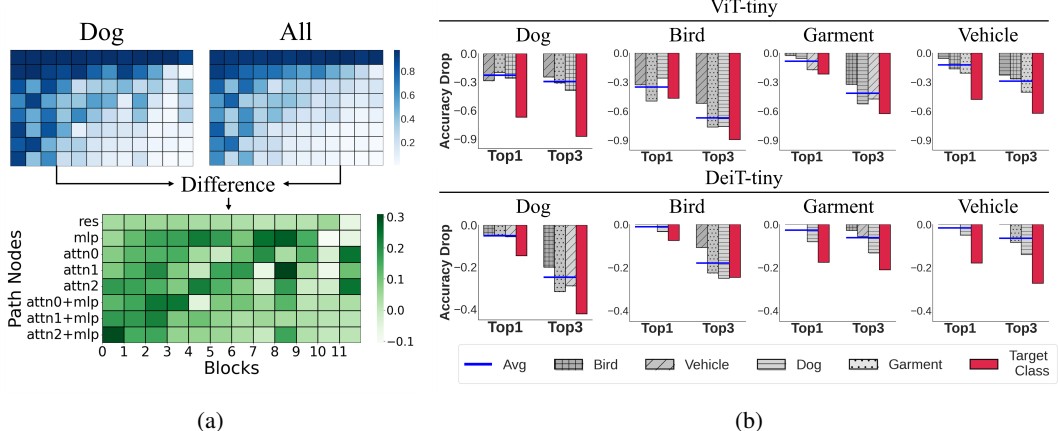

(a)                                                                        (b)

Figure 4: **Causal paths uniquely activated for specific classes. (a)** Average causal path ratios for a target class (left), all classes (right), and their difference (bottom), highlighting class-specific paths. Here, res, mlp, and attn# indicate residual, MLP, and attention paths from head #, respectively. **(b)** Accuracy drop when ablating the most class-specific path, showing selective reliance by each class.

We find that self-repair occurs less frequently on the causal path. While self-repair is known to be highly noisy, as noted in [16], the results still show a clear difference: both the mean and median scores are consistently lower on the causal path than off it. This suggests that the causal path captures components essential to the decision and less reliant on backup mechanisms. In other words, the selected paths carry information not easily replaceable, underscoring their critical role for decision.

**Class-specific causal subpaths play a functional role in predicting their respective classes**    Here, we aim to investigate whether the discovered causal paths contain class-wise causal nodes—nodes that are consistently utilized across samples within the same class group—and whether these nodes play a significant role in the model's classification decisions. To improve clarity, we first select four super-classes—dog, bird, garment, and vehicle—among the 1,000 ImageNet classes based on semantic similarity derived from WordNet. We then aggregated the causal paths extracted from individual samples and compiled statistics on the frequency of each subpaths' occurrence. By comparing these frequencies to the overall average across all samples, we identified causal subpaths that were significantly more active within specific super-classes (as shown in Figure 4-(a)). We refer to these as class-wise causal subpaths, hypothesizing that *they store key discriminative information relevant to their respective super-classes* due to their unusually high activation rates.

To validate this hypothesis, we intervene in the class-wise causal subpaths and measure the performance drop. If these nodes indeed encode class-specific information, their removal should lead to a greater accuracy drop within the corresponding super-class than in others. Figure 4-(b) clearly demonstrates this pattern. For instance, when the class-wise causal subpaths for the dog super-class were deactivated in a ViT-tiny model, the top-1 accuracy for dog samples decreased by approximately $44.7\%$ more than that for other super-classes. Similar trends were observed across bird, garment, and vehicle classes, indicating that the proposed metric functions consistently across the model.

It is important to note that due to inherent semantic overlap among ImageNet classes, interventions on class-wise causal subpaths may still affect the logits of unrelated classes. Additionally, due to

|  | Hit. ($\uparrow$) | Faith. ($\uparrow$) | Spars. ($\downarrow$) |  |  | Hit. ($\uparrow$) | Faith. ($\uparrow$) | Spars. ($\downarrow$) |
|---|---|---|---|---|---|---|---|---|
| $NT_1$ | 0.0000 | 0.0005 | 0.6571 |  | $NT_1$ | 0.0105 | 0.0136 | 0.7276 |
| $NT_{10\%}$ | 0.0000 | 0.0006 | 0.5648 |  | $NT_{10\%}$ | 0.0078 | 0.0133 | 0.0799 |
| $ET_{all}$ | 0.2079 | 0.2354 | 0.9806 |  | $ET_{all}$ | 0.4454 | 0.3166 | 0.9999 |
| $ET_{cls}$ | 0.4808 | 0.4734 | 0.9909 |  | $ET_{cls}$ | 0.2627 | 0.1832 | 0.9650 |
| **CPT** | **0.9826** | **0.5466** | **0.8641** |  | **CPT** | **0.9638** | **0.2991** | **0.7280** |

Table 2: **Quantitative results (language).** Averaged over three models on two datasets; full results in Appendix.

Table 3: **Quantitative results (vision).** Averaged over two models on two datasets; see Appendix for details.

visual diversity within each super-class, turning off only a small number of subpaths may not entirely collapse performance. Nevertheless, the consistent and pronounced patterns observed across all super-classes suggest that our method effectively identifies causal subpaths that play a meaningful role in class-specific inference.

**Quantitative results show our method yields reliable and faithful explanations**  Each value in Tables 2 and 3 represents the average score across models on two datasets. All methods are evaluated by pruning the model to retain only the paths identified by each method. We report three metrics: **Hit.** (hit rate) measures the proportion of cases in which the pruned model produces the same decision as the original; **Faith.** (faithfulness) quantifies the ratio of the original logit preserved after pruning; and **Spars.** (sparsity) denotes the proportion of model parameters retained by the identified path.

Our method (CPT) achieves a near-perfect hit rate, consistent with the theoretical guarantee in Theorem 2 that the identified paths are reliably causal. In contrast, existing methods show substantially lower hit rates, supporting our claim in Table 1 that while tracing is feasible with backward chaining, it is generally not reliable for identifying true decision paths. CPT also achieves the highest faithfulness, indicating that it preserves the model's original decision behavior more accurately than alternative methods. Notably, it does so while retaining significantly fewer parameters: whereas edge-level methods such as $ET_{all}$ and $ET_{cls}$ rely on nearly the entire model, CPT produces more faithful and compact explanations through substantially more efficient path selection.

## 4   Conclusion

In this paper, we presented an automated framework for tracing causal paths given a decision. We provide both theoretical analysis and empirical evidence showing that our method efficiently uncovers all causal paths responsible for a decision, with average-case polynomial-time complexity. Furthermore, we demonstrated that the identified causal paths (1) are less susceptible to self-repair effects, (2) reveal the structural grounds for subpaths uniquely activated for specific classes, and (3) yield more faithful and precise explanations than existing methods.

**Limitations and Future Work.**   First, the identified causal paths are derived under the assumptions of our proposed framework and may not generalize under different assumptions. In particular, our unfolding procedure assumes uniform propagation of bias terms across all paths; however, accurately quantifying their individual contributions is non-trivial and remains an open direction for future work. Second, we acknowledge that our experiments were conducted on smaller models compared to state-of-the-art architectures. Although our method achieves polynomial-time complexity on average, large models may still incur prohibitive runtime in worst-case scenarios, and the reduced search space can remain sizable. Extending our minimality-based subset search to also prune supersets of non-causal subsets could mitigate this issue. Lastly, while our analysis focuses on structural mechanisms within the model, it opens avenues for future integration with feature attribution methods, potentially bridging structural and feature-level interpretability.

Despite these limitations, our work is the first to propose an efficient and reliable framework for tracing causal paths within transformer models for a given decision. We believe this represents an important step toward making transformers more transparent and robust in safety-critical domains, helping to prevent misuse and improve trust in deployment.

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
