# OpenReview forum: "Causal Path Tracing in Transformers"
_NeurIPS.cc/2025/Conference — Submitted to NeurIPS 2025_

### Official Review · Reviewer_3ZGp · 2025-06-19

**Clarity:** 2
**Significance:** 3
**Originality:** 2
**Rating:** 4
**Confidence:** 3

**Summary:**

The paper proposes an automated framework for improving the interpretability of transformer models. Specifically, the authors aim to identify internal components that causally influence a given decision made by the model. To achieve this, they model the transformer as a causal graph, where computational components are represented as nodes and their dependencies as edges. Within this framework, they seek to identify minimal sets of nodes responsible for a particular decision by applying interventions. The results demonstrate the effectiveness of the proposed method on both language and vision datasets.

**Questions:**

1. What is the actual wall clock time (with standard deviation or error bars) required to identify causal paths per decision, as shown in Figure 2?
2. Could the authors clearly define the following terms in the main paper?
    - Interpretability in Condition 5.c
    - Reliability of a causal node set
    - Computational dependency and edge definition in the causal graph
    - What exactly are "alternative token embeddings" used in TOKEN RESAMPLING? Are they from other training samples, or randomly sampled from the embedding matrix?

I liked the paper and I am willing to raise my score if authors can clarify my questions.

**Ethical Concerns:**

["NO or VERY MINOR ethics concerns only"]

**Final Justification:**

The rebuttal clarified key definitions which I found helpful and recommend including in the paper. While the idea appears promising, several technical aspects fall outside my expertise, and I cannot fully assess their significance or correctness. I have increased my score to Borderline Accept based on the improved clarity and potential contribution, but I defer to other reviewers and the AC for a deeper evaluation.

**Limitations:**

Yes

**Quality:**

2

**Strengths And Weaknesses:**

Strengths:
+ The paper addresses a highly relevant problem and offers an interesting and principled solution.
+ Extensive empirical evaluation across both language and vision models. The experiment analyzing the effectiveness of class-specific causal paths (Figure 4) is particularly insightful and well-motivated.

Weaknesses [please see details]:
- Lacks and missing key definitions
- Limited novelty

Details:
I appreciate the motivation behind the paper and the direction the authors toke in addressing a timely and important problem. However, as I read through the paper, I found myself increasingly confused rather than excited. Several terms and claims are introduced without sufficient explanation or justification.

For instance, Condition 5.c is central to the proposed notion of a "sufficient intervention" yet the term interpretable is used without a precise or quantifiable definition. What exactly does it mean for an intervention to be interpretable in this context? Similarly, the authors claim that TOKEN RESAMPLING satisfies the sufficient intervention conditions. However, it is not clear how. The paper mentions resampling from "alternative token embeddings" but what exactly are these embeddings? Are they from different tokens in the vocabulary? Do they require modifications to the model architecture to be used effectively?

In addition, several other concepts are either underdefined or entirely missing. For example:
- What exactly constitutes reliability in the context of a causal node set?
- What is the precise definition of computational dependency—does it refer to the relationship between modules within a block (e.g., attention $\rightarrow$ MLP)?
- How is the edge between two nodes defined concretely in the causal graph?

While I can infer some of these definitions based on context and prior work, the paper should provide formal definitions to ensure clarity and reproducibility.

My second concern relates to the novelty of the technical contributions. One of the central ideas, that average-case polynomial-time search is achievable, is a relatively direct application of the causal minimality condition (Definition 4.c), based on the Halpern–Pearl framework. Given this, the proposed search algorithm seems like a natural extension rather than a fundamentally new technique.

Minor Comments:
- Figure 3 is not referenced anywhere in the main text. This should be corrected.

---

> ### Author Rebuttal · Authors · 2025-07-31
>
> We sincerely thank the reviewer for the valuable and constructive feedback, which helped improve the clarity and rigor of our paper.
>
> ### **[R1] Clarifying Definition 5.c**
>
> The reviewer highlights the ambiguity in Definition 5.c, particularly regarding the term *"interpretable."* We agree this phrasing lacks precision and will revise the definition to clarify our intended constraint.
>
> Our goal in Definition 5.c is to ensure interventions don't rely on out-of-distribution (OOD) representations. In causal inference, an intervention typically means forcibly assigning a value that "*could have occurred in the original system"* [1]. In our context, the “original system” corresponds to the manifold of representations that the model can naturally generate using its learned parameters of that specific layer. Thus, using OOD representations may violate this assumption and leads to confounded interpretations, either falsely attributing causal effects or failing to detect genuine ones.
>
> To address this concern, we will revise Definition 5.c as follows:
>
> (5.c) Intervention Controllability.
>
> The intervention must replace a representation with one that is drawn from the model’s in-distribution at the corresponding layer. In other words, although the value is forcibly assigned, it must remain on the representation manifold learned by the model, to avoid confounding effects from off-distribution artifacts or unknown factors.
>
> [1] Judea Pearl. Causality: Models, Reasoning and Inference.
>
> ***
>
> ### **[R2] Clarifying How TOKEN RESAMPLING Works**
>
> TOKEN RESAMPLING is an intervention strategy that replaces a hidden representation at a given layer with **that of different tokens from the vocabulary, specifically ones that were not used during that inference.** We precompute and cache these representations under the same configuration; no architectural changes are required.
>
> ***
>
> ### **[R3] About Intervention**
>
> To illustrate the distinctions between intervention methods, we consider a simplified transformer architecture with a single block and a single attention head; thus, the block input $z_{ib}$ corresponds to the original token embedding $z_{token}$. The block output can then be decomposed into four additive path terms, as described in Equation (2) of the main paper: the residual only path node ($v_1$), the Attention only path node ($v_2$), Attention+MLP path node ($v_3$), and MLP only path node($v_4$).
>
> **[R3-1] For Definition 5.a**
>
> Consider two intervention scenarios where either $v_1$ or $v_4$ is replaced.
>
> Under TOKEN RESAMPLING, we substitute the original token embedding $z_{token}$ with a different in-vocabulary embedding $z_{token}'$. This substitution affects both $v_1$ and $v_4$, as they both contain $z_{token}$ (i.e., $z_{ib}$) as their root variable. Crucially, whether $z_{token}$ or $z_{token}'$ appears in $v_1$ or $v_4$ leads to distinct expressions for the block output. By these expressions, the resulting interventional graphs remain distinguishable across scenarios. This one-to-one correspondence between the mathematical form and the graph structure holds not only in this toy example but also more generally, thereby satisfying **Definition 5.a.**
>
> In contrast, DIRECT NOISE adds the same noise directly to multiple nodes (e.g., $v_1$, $v_4$), which leads to overlapping outcomes across different intervention scenarios. This is due to the additive nature of the block output, where all path node contributions are combined through addition. As a result, this destroys the one-to-one mapping between graphs and their formulas, thereby violating **Definition 5.a.**
>
> **[R3-2] For Definition 5.b**
>
> Consider an intervention scenario where all path nodes are replaced.
>
> Under ZERO MASKING, each component is zeroed out, which causes the model output to collapse to a constant vector (i.e., the block output $z_{ob} = \mathcal{0}$), resulting in a fixed decision $y = b_{cls}$, where $b_{cls}$ denotes the classifier's bias term. In this case, the intervened decision becomes entirely independent of the original input, relying solely on the classifier bias. As a result, there exists at least one decision for which the causal necessity condition is never satisfied, thereby violating **Definition 5.b.**
>
> In contrast, TOKEN RESAMPLING prevents the decision from collapsing to the classifier bias $b_{cls}$ by replacing the original token embedding $z_{token}$ with a different in-vocabulary embedding $z_{token}'$, ensuring that the causal necessity condition can still be satisfied and thus upholding **Definition 5.b.**
>
> **[R3-3] For Definition 5.c**
>
> Consider an intervention scenario where $v_4$ is replaced.
>
> TOKEN RESAMPLING replaces a representation with another token embedding from the model's learned vocabulary. Because these embeddings originate from the model itself, the resulting representations remain on the model's manifold, ensuring that interventions stay within the model's natural distribution and thus satisfying **Definition 5.c.**
>
> In contrast, NOISE TOKEN replaces $z_{token}$ with $z_{token} + z_{noise}$, where $z_{noise}$ is the noise. As this composite input is passed through the MLP in $v_4$, the noise term is transformed by learned parameters, producing an activation that is not guaranteed to lie on the model's representation manifold. Furthermore, as this noisy signal propagates through subsequent layers, this may lead to out-of-distribution representations, thereby violating **Definition 5.c.**
>
> ***
>
> ### **[R4] Clarifying Other Concepts**
>
> **[R4-1] Reliability**
>
> Causal evaluation reliability is defined as the proportion of considered paths to all possible decision paths. An evaluation that accounts for all possible paths (or is theoretically guaranteed to do so) has full reliability (i.e., 1).
>
> **[R4-2] Computational Dependency (with Edge)**
>
> A *computational dependency* is defined as a transformation relationship between two nodes. If the transformation is atomic, i.e. not decomposable into intermediate transformations, it is referred to as a *direct computational dependency*. Here, direct dependencies correspond to *edges* in the graph.
>
> For example, if nodes are set at the feature level, such as the input and output features of a single linear layer, the affine transformation by the layer constitutes a computational dependency between the two nodes. However, since it can be decomposed into a weight multiplication and bias addition, it is not considered direct. In this case, an intermediate feature-level node lies between the two.
>
> As another example, if nodes are set at the path level, as in our method, the summation constitutes a computational dependency between path nodes and the block output; this dependency is direct.
>
> ***
>
> ### **[R5] Clarifying Novelty**
>
> We would like to clarify that while our method builds on the Halpern–Pearl (HP) framework, our core contributions go significantly beyond it. The HP framework provides a criterion for what is a cause, but it does not address how to efficiently identify such causes in complex models like transformers.
>
> Even when adopting the HP criterion, finding causal sets among all parent subsets is computationally intractable due to the need to evaluate exponentially many combinations, making the problem NP-complete. The HP framework offers no algorithmic strategy to overcome this.
>
> Moreover, when multiple causal sets are found within a block, evaluating their up/downstream impact becomes even more intractable, as each set must be treated independently. This again leads to exponential complexity, making end-to-end causal tracing infeasible.
>
> In contrast, our work makes two distinct contributions that enable practical and efficient causal evaluation:
>
> - **Efficient within-block search**: By leveraging the minimality condition, we introduce a new algorithm that prunes the exponential search space through reordering the search space. This is not a heuristic—we provide a theoretical proof that our algorithm runs in average-case polynomial time, which is a core contribution.
>
> - **Scalable block-wise composition**: We extend causal evaluation across blocks without incurring exponential overhead. Crucially, we provide a theoretical guarantee that full reliability is preserved under this composition, enabling end-to-end tracing.
>
> These aspects are essential for causal tracing in transformers. Without them, adopting any causal criterion still leaves the computational problem intractable. As summarized in Table 1, no prior work has provided both theoretical tractability and empirical coverage of full transformer internals. Our method is the first to offer both, representing a novel contribution.
>
> Additionally, our method helps illustrate internal transformer operations by unfolding block-level computations into meaningful causal paths, which holds interpretability value in its own right.
>
> ***
>
> ### **[R6] Actual Wall-Clock Time**
>
> ||Total Search Space|$\text{NT}$ (<0.0001%)|$\text{ET}$ (<0.0001%)|$\text{CPT}$ (100%)|
> |-|-|-|-|-|
> |Pythia-1b|$2^{10 \times 16}$|1.31sec ±0.17|8.49sec ±0.36|3114.00sec ±930.95|
> |Pythia-14m|$2^{6 \times 6}$|0.13sec ±0.01|1.76sec ±0.01|10.04sec ±0.86|
>
> The table shows the actual wall-clock time per sample for each method. The “Total Search Space” column represents all possible combinations of internal path nodes across layers, computed as $2^{n \times D}$, where $n$ is the number of path per block and $D$ is the number of blocks. The percentages in parentheses indicate each method’s coverage over the total search space.
>
> Notably, prior methods such as $\text{NT}$ and $\text{ET}$ cover less than 0.0001% of the space, as they do not consider "combinations" of internal path nodes. In contrast, **our method achieves full coverage (100%) by evaluating these combinations, yet remains computationally feasible with the proposed pruning strategies, showing reasonable runtimes even compared to other methods with drastically smaller coverage.**

---

> > ### Comment · Reviewer_3ZGp · 2025-08-05
> >
> > I thank the authors for their response. I found the clarification around the definition helpful, and I would suggest including that clarification in the final version of the paper.
> >
> > The argument about the use of the minimality heuristic to explore a large search space more efficiently is interesting. While I see the potential value, I am not entirely confident in assessing the technical soundness or the broader significance of this approach, as some aspects of the rebuttal go beyond my expertise.
> >
> > Given this, I am increasing my score to Borderline Accept to reflect the improved clarity and the potential for contribution, but I am also lowering my confidence and defer to the other reviewers and area chair for a more thorough technical evaluation.

---

### Official Review · Reviewer_s7J4 · 2025-06-23

**Clarity:** 3
**Significance:** 3
**Originality:** 3
**Rating:** 4
**Confidence:** 1

**Summary:**

This paper introduces a causal path tracing framework to analyze how information flows through the internal components of a Transformer model to make a specific decision. The approach unfolds each Transformer block into a causal graph of path nodes and applies a minimality-based subset search to identify all possible causal paths within each block, achieving polynomial-time complexity on average. To enhance reliability, the authors also propose a union-based causal path reference strategy, allowing for efficient and faithful tracing of causal influence across the model.

**Questions:**

Q1: The paper defines a “model decision” in the context of causal path tracing. Is it the final prediction (e.g., classification label), a logit score, or something else? Moreover, in the case of generative Transformer models (e.g., language models), which produce multiple sequential outputs, how does the framework handle such decisions? Can the proposed causal graph structure and path tracing procedure be extended or adapted to trace information flow across multiple output steps?

Q2: While the paper demonstrates that the traced causal paths are faithful and reveal interesting internal behavior, it is still not entirely clear what practical use cases this method enables. Could the authors elaborate on how this framework can help in real-world applications?

**Ethical Concerns:**

["NO or VERY MINOR ethics concerns only"]

**Final Justification:**

I am not an expert in this domain. I have tried to understand this paper and discuss with the authors. Based on the fact that I have little knowledge about this domain, I adjust my confidence from 2 to 1.

**Limitations:**

See Questions.

**Quality:**

3

**Strengths And Weaknesses:**

**Disclaimer: I am not an expert in this specific field, and my review reflects a general understanding based on my background. I may not be qualified to fully assess the technical novelty or correctness of this submission.**

Strengths:
* The paper is generally well-written and articulates a clear motivation for understanding the internal decision process of transformer models through causal analysis.
* Introducing a causal graph framework and minimality-based path search offers a relatively unique lens for interpreting Transformer behavior
* The authors emphasize both theoretical grounding (e.g., polynomial-time complexity) and automation, which adds practicality to their proposed approach.
* The paper includes experimental results showing interesting phenomena (e.g., class-specific path activation, reduced self-repair effects along causal paths), which adds interpretive insights even if the metrics are somewhat qualitative.

Weaknesses:
* It remains unclear how the proposed tracing method could directly assist in downstream tasks (e.g., model debugging, robustness improvement, or fairness). The current experiments focus more on interpretability phenomena than on practical impact.

---

> ### Author Rebuttal · Authors · 2025-07-31
>
> We sincerely thank the reviewer for the valuable and constructive feedback, which helped improve the clarity and rigor of our paper.
>
> ### **[R1] Practical application in downstream tasks**
>
> While our experiments primarily focus on interpretability-oriented phenomena (e.g., identifying causal paths), **the proposed algorithm can be naturally extended to practical downstream tasks such as model debugging and pruning.** For example, by tracing the causal path of a misclassified sample, one can inspect where the model relies on non-causal components to make incorrect decisions. Moreover, pruning nodes that are not included in the causal paths of given samples—either during inference or fine-tuning—has the potential to reduce model size and computational cost without harming decision quality.
>
> ***
>
> ### **[R2] Clarifying "model decision"**
>
> In our response, the term "model decision" specifically refers to the model's final prediction for a single output token. For clarity, we have formally defined this notion in Definition 2 of the manuscript.
>
> ***
>
> ### **[R3] Extended to multiple sequential output cases**
>
> In line with prior research (e.g., [1,2]), our algorithm primarily focuses on classification and single-token generation tasks. A key challenge in extending causal path analysis to complex generative tasks, such as multi-token generation, is the potential existence of multiple causal paths, along with the need to understand their interactions. Accordingly, we leave the question of how to identify and unify causal paths for multi-token generation as an open direction for future work. Addressing this challenge could lay the groundwork for training models that better adhere to causal reasoning, while also supporting more systematic auditing and refinement of non-causal decision mechanisms.
>
> [1] Kevin Meng, David Bau, Alex Andonian, and Yonatan Belinkov. Locating and editing factual associations in gpt.
>
> [2] Aaquib Syed, Can Rager, and Arthur Conmy. Attribution patching outperforms automated circuit discovery.

---

> > ### Comment · Reviewer_s7J4 · 2025-08-05
> > **Response to the Authors**
> >
> > Thank you for your response. The authors has addressed my concerns. I keep my score.

---

### Official Review · Reviewer_KqMq · 2025-06-23

**Clarity:** 3
**Significance:** 3
**Originality:** 3
**Rating:** 4
**Confidence:** 3

**Summary:**

The paper proposes a new mechanism for discovering causal paths for transformers to explain decisions. The proposed mechanism unfolds transformer blocks into path nodes and applies minimality-based causal subset search over all blocks to discover the causal path for a given decision. The paper evaluated the proposed method and showed that it outperforms baselines while being efficient. The paper also demonstrated that the discovered causal paths are important for the final decisions.

**Questions:**

See W1 above.

**Ethical Concerns:**

["NO or VERY MINOR ethics concerns only"]

**Final Justification:**

I find the idea of unfolding transformer blocks into nodes and using minimality search to uncover causal paths technically compelling. The authors demonstrate both the soundness of their approach and its effectiveness on classification tasks through extensive experimental validation. Given the technical merit and solid evidence, I believe the paper merits a score of 4.

However, I cannot justify a higher rating due to the limited downstream applications. Most LLM use cases involve long-sequence generative tasks, yet there remains no satisfactory method for explainability in these scenarios (particularly from the causality angle). The impact of this work would be more convincing if the authors could demonstrate how their method extends to such timely and crucial tasks. The authors note this as possible future work, and I understand the time constraints during the rebuttal period. Without additional results, I will keep my rating unchanged.

**Limitations:**

Yes

**Quality:**

3

**Strengths And Weaknesses:**

S1: The approach of unfolding transformer blocks into nodes and applying the minimality search to discover the causal paths is sound and effective.

S2: The causal paths discovered by the proposed approach are shown to be important.

S3: The paper is well-structured and easy to follow.

W1: The discovered causal paths are for classification tasks. The proposed approach could have a higher impact if the authors could show how the proposed approach can help discover causal paths to explain more complex generative tasks (e.g., hallucinations and logical fallacies when solving math problems).

---

> ### Author Rebuttal · Authors · 2025-07-31
>
> We sincerely thank the reviewer for the valuable and constructive feedback, which helped improve the clarity and rigor of our paper.
>
> ### **[R1] Extension beyond classification: applicability to complex generative tasks**
>
> In line with prior research (e.g., [1,2]), our algorithm primarily focuses on classification and single-token generation tasks. A key challenge in extending causal path analysis to complex generative tasks, such as multi-token generation, is the potential existence of multiple causal paths, along with the need to understand their interactions. Accordingly, we leave the question of how to identify and unify causal paths for multi-token generation as an open direction for future work. Addressing this challenge could lay the groundwork for training models that better adhere to causal reasoning, while also supporting more systematic auditing and refinement of non-causal decision mechanisms.
>
> [1] Kevin Meng, David Bau, Alex Andonian, and Yonatan Belinkov. Locating and editing factual associations in gpt.
>
> [2] Aaquib Syed, Can Rager, and Arthur Conmy. Attribution patching outperforms automated circuit discovery.

---

> > ### Comment · Reviewer_KqMq · 2025-08-05
> >
> > Thank you for your response. My concern is addressed, so I keep my rating the same.

---

### Official Review · Reviewer_6s2J · 2025-07-01

**Clarity:** 1
**Significance:** 2
**Originality:** 3
**Rating:** 3
**Confidence:** 4

**Summary:**

This paper addresses the problem of extracting a faithful subgraph of a transformer that explains the model's computation on some input set (circuit discovery). The paper proposes a novel method, with perhaps the most important innovation a minimality-based subset search that keeps the layer-wise search for component subsets more manageable.

**Questions:**

* line 146: “we apply a similar simplification to layer normalization” this is potentially not faithful – isn’t this at odds with the claim that the method is guaranteed to be faithful? [made e.g. in line 286 "uncovers all causal paths"]
* line 71: is there any constraint on how the directions of the edges must respect the ordering of the node sets?
* Definition 3 seems to define “path” as a sequence of node sets. Is this really a nratual notion of “path”?

Minor:
* minor: (line 164) “cannot be solved”  assuming P != NP

**Ethical Concerns:**

["NO or VERY MINOR ethics concerns only"]

**Final Justification:**

The submitted manuscript had major clarity issues. The authors have extensively provided clarifications in the rebuttal, which I appreciate very much. Incorporating these will undoubtedly improve the paper a lot.
Nonetheless, I am not in support of accepting. Given the poor level of clarity of the initial writeup, I think another round of review would be needed for fully assessing the validity of the paper, and would be appropriate before acceptance at a high-quality conference.

I have increased my score to reflect the fact that I do believe the paper will be substantially clearer with the promised changes, but am not in support of accepting.

**Limitations:**

yes

**Quality:**

2

**Strengths And Weaknesses:**

Strengths:
* addresses a foundational problem of high current interest in the field (circuit discovery)
* proposes an efficient solution to the layer-wise search for subsets of components

Weaknesses:

The major weakness of the paper is in the clarity of writing. Even though I am familiar with the circuit discovery literature, I found it hard and effortful to understand the proposed method.
* (A) Clarity issues in the presentation of the method itself, which impede understanding the method itself: There seems to be a type mismatch in Algorithm 2. In line 2, \mathcal{P} is defined as a length-D sequence of node sets. On the other hand, line 9 updates \mathcal{P} by unioning it with a set of paths. Also, Algorithm 2 just outputs a single path in the sense of Definition 3, whereas the paper in various places speaks of discovering multiple paths.
* (B) Clarity issues in other places of the paper, which impede following the paper. I’m listing a few specific places, which mostly involve technical-sounding statements that are not defined in the paper:
* * line 42: “referencing the union of causal paths” → at this point it’s unclear to the reader what this means
* * Definition 5.c “must remain interpretable, rather than being overwhelmed by the parameters of the intervention” – this is not formal and I don’t understand what it means. As a consequence, it is hard to judge whether Definition 5 applies in a given case or not.
* * line 124: “distorting Property 1” why?
* * line 152: “grouped into distinct paths”: It wasn’t clear when reading how exactly this is done
* * line 196: “causal subpath reference” – I didn’t see this term defined further up
* * line 209: “all baselines are extended under a backward chaining framework” what does this mean
* * line 219: “enumerates all decision paths for a given output”: the paper doesn’t seem to define what this means. Algorithm 2 outputs a single path \mathcal{P} (a sequence of node sets) by Definition 3. In what sense does the method then enumerate all paths?
* * line 292: “uniform bias propagation across all paths” what does this mean?

---

> ### Author Rebuttal · Authors · 2025-07-31
>
> We sincerely thank the reviewer for the valuable and constructive feedback, which helped improve the clarity and rigor of our paper.
>
> ### **[R1] Clarifying Concern (A)**
>
> **[R1-1] For the definition of causal path and causal referencing**
>
> In Definition 3, **our intention is that the term 'causal path's represents the set of traced causal paths, not a single path.** That is, the set $\mathcal{P}$ can contain multiple causal paths. The reason is as follows: each causal node set $V_i \in \mathcal{P}$, when paired with its corresponding subpath reference $P$ (see the definition below), satisfies all three conditions (4.a), (4.b), and (4.c) in Definition 4. In other words, each subset $V_i \cup P$, a subset of $\mathcal{P}$, is itself a single causal path that is sufficient to produce the original decision.
>
> **Definition.** Given a node set $V$, we define a subpath reference as a collection of consecutively connected downstream node sets that serve to bridge $V$ to the output. If all
> such downstream node sets are causal node sets (i.e., satisfy Definition 4), we refer to the collection as a causal subpath reference. Thus, referencing for $V$ means considering all possible subpath references of $V$, and causal referencing for $V$ means considering all possible causal references of $V$.
>
> **Remark.** Without *causal referencing*, one cannot guarantee that a identified node set is a true cause of the model’s decision. This is because only the full collection of downstream node sets is considered when evaluating causal effect.
>
> For example, consider a two-block transformer in which two node sets $V_{1}^{(2)}$ and $V_{2}^{(2)}$ in the final block have already been identified as causal node sets, respectively. In this case, whether a node set $V_1^{(1)}$ in the first block is causal may depend on whether it is evaluated jointly with $V_{1}^{(2)}, V_{2}^{(2)}$, or both (the case of neither can be safely ignored due to the causal edge property). That is, the causal status of $V_1^{(1)}$ can vary depending on which downstream path the internal information propagates through; in other words, some combinations may be sufficient for the decision, while others may not.
>
> **[R1-2] For the algorithm**
>
> Building on the above clarification, Algorithm 2 was intended to describe how causal paths are constructed by iteratively collecting per-block causal node sets into $\mathcal{P}$. To avoid confusion, we will revise the algorithm to explicitly use a separate variable for causal node sets per block and reserve $\mathcal{P}$ for the final collection after traversing all blocks.
>
> Furthermore, in Algorithm 2, based on the theoretical justification provided by Theorem 2, **we treated multiple causal paths as a single causal path during causal referencing.** This is why, in other parts of the paper, we refer to them as multiple causal paths. Specifically, Theorem 2 proves that the union of block-wise discovered causal node sets can be used as a causal subpath reference. This means that multiple downstream causal paths (more precisely, causal “subpaths” from intermediate nodes to the output) can be referenced collectively as a single path. In this sense, indicating *multiple causal paths* was intended to emphasize the multiplicity of downstream causal subpaths that contribute to the decision. To avoid further confusion, we will add a clarifying explanation in the main paper to make this interpretation explicit.
>
> ***
>
> ### **[R2] Clarifying Concern (B)**
>
> **[R2-1]**
> - L42: “referencing the union of causal paths”
> - L196: “causal subpath reference”
>
> We hope that the clarification in Concern (A), which defines referencing and explains how Theorem 2 justifies referencing the union of causal paths, adequately addresses this point. As noted, we will revise  the phrase to "causal subpaths" here as well, in order to avoid confusion.
>
> **[R2-2]**
> - L124: “distorting Property 1”
>
> To illustrate the distinctions between intervention methods, we consider a simplified transformer architecture with a single block and a single attention head; thus, the block input $z_{ib}$ corresponds to the original token embedding $z_{token}$. The block output can then be decomposed into four additive path terms, as described in Equation (2) of the main paper: the residual only path node ($v_1$), the Attention only path node ($v_2$), Attention+MLP path node ($v_3$), and MLP only path node($v_4$).
>
> Consider an intervention scenario where all path nodes are replaced.
>
> Under ZERO MASKING, each component is zeroed out, which causes the model output to collapse to a constant vector (i.e., the block output $z_{ob} = \mathcal{0}$), resulting in a fixed decision $y = b_{cls}$, where $b_{cls}$ denotes the classifier's bias term. In this case, the intervened decision becomes entirely independent of the original input, relying solely on the classifier bias. As a result, there exists at least one decision for which the causal necessity condition is never satisfied, thereby violating **Definition 5.b.**
>
> In contrast, TOKEN RESAMPLING prevents the decision from collapsing to the classifier bias $b_{cls}$ by replacing the original token embedding $z_{token}$ with a different in-vocabulary embedding $z_{token}'$, ensuring that the causal necessity condition can still be satisfied and thus upholding **Definition 5.b.**
>
> **[R2-3]**
> - L152: “grouped into distinct paths”
>
> **Each term in Equation (2) can be categorized based on the layer parameters involved, attention, MLP, or none.** Taking the block input $z_{ib}$ as the root, the term unaffected by any parameters represents the residual-only path, while the others are classified according to the parameters they interact with. Note that we did not explicitly expand intermediate terms such as $z_{q}^{(h)}$ in terms of $z_{ib}$, as doing so would make the equation unnecessarily long; however, we would like to clarify that these are linear projections of $z_{ib}$ within the attention layer.
>
> **[R2-4]**
> - L209: “all baselines are extended under a backward chaining framework”
>
> This reflects what we remarked in Concern (A): the baselines trace from the decision to each block **without causal referencing and simply concatenate the results (i.e., this is what we mean by backward chaining).** This is because no prior work has addressed the intractable complexity that arises from implementing causal referencing, and thus each baseline could only be implemented in the way originally proposed.
>
> **[R2-5]**
> - L219: “enumerates all decision paths for a given output”
>
> We hope that our revision of the definition of causal paths in Concern (A), along with our clarification regarding the meaning of indicating multiple causal paths, sufficiently addresses this point. That is, while identifying causal paths by considering all possible paths contributing to a decision was infeasible in prior works, our method opens the door to making this process tractable. We would appreciate it if this key contribution were acknowledged.
>
> **[R2-6]**
> - L146: “we apply a similar simplification to layer normalization”
> - L292: “uniform bias propagation across all paths”
>
> As noted in the main paper, **this reflects limitations of our unfolding assumption; nevertheless, we believe our method remains significant, as demonstrated by its empirical performance:** the hit rate, measuring whether the pruned model preserves the original decision, consistently approaches 1.0.
>
> Specifically, layer normalization introduces non-linearity (like softmax or GeLU), in the sense that its internal parameters (mean and variance) depend on each input. This makes it difficult to analytically unfold the block output solely in terms of the block input. A fully rigorous treatment would require performing causal evaluation at every computation within a block, which would further render the evaluation process intractable. To make the analysis tractable, we adopt a computational trick by precomputing the input-dependent statistics and treating them as fixed constants, thereby approximately linearizing the normalization operation.
>
> Similarly, for bias terms, although in principle each attention head could receive a different degree of contribution from the bias, we approximate this by distributing the bias equally across all head-specific paths, for the same tractability reason.
>
> Importantly, we confirm that these approximations do not significantly affect the quality of the causal evaluation, as supported by our empirical results. That said, we acknowledge that resolving these limitations remains an important direction for future work. We will include an explanation of this in the main paper.
>
> **[R2-7]**
> - L71: for the directions of the edges
>
> Our goal is to identify the paths along which influence flows from the input to the final decision, even though the tracing procedure itself operates in a backward manner. Accordingly, we impose a constraint that **all edges in the causal graph follow the forward computational direction of the model.** We will include this clarification to make the construction of the causal graph more precise.
>
> **[R2-8]**
> - For the notion of “path”
>
> Similar to the definition of subpath reference in Concern (A), our use of the term “path” as a sequence was intended to represent a collection of consecutively connected node sets; specifically, **a sequence in which each node set receives incoming edges from exactly one preceding node set (or at minimum, from the model input) and passes outgoing edges to exactly one subsequent node set (or at minimum, to the model output) within the collection.** To avoid further confusion, we will include a more precise explanation of this notion of a path in the main paper.
>
> **[R2-9]**
> - L164: “cannot be solved”
>
> We will revise this expression to avoid redundancy. Thank you for pointing it out.

---

> > ### Comment · Reviewer_6s2J · 2025-08-04
> >
> > Thanks to the authors for clarifying these points!
> >
> > Incorporating these clarifications into the paper will help improve it a lot.

---

> ### Comment · Reviewer_6s2J · 2025-08-04
>
> Can the authors briefly explain how this question here is addressed?
>
> > There seems to be a type mismatch in Algorithm 2. In line 2, \mathcal{P} is defined as a length-D sequence of node sets. On the other hand, line 9 updates \mathcal{P} by unioning it with a set of paths
>
> I tried understanding the detailed explanations, which are appreciated, but would also appreciate a to-the-point answer to this question.

---

> > ### Author Response · Authors · 2025-08-05
> >
> > Thank you for the follow-up. To clarify, the type mismatch arose because $\mathcal{P}$ was originally used as a sequence, but in Line 9 of Algorithm 2, it was updated using a set union operator ($\cup$). We resolve this by replacing $\cup$ with a sequence concatenation operator as follows:
> >
> > $\text{Line 9}: \mathcal{P} \leftarrow \mathcal{P} + [V_{\text{out}}^{(j)}]; \quad \mathcal{G}_c \leftarrow \text{block } j \quad \triangleright \text{Append to sequence (concatenation)}$

---

> > > ### Comment · Reviewer_6s2J · 2025-08-05
> > >
> > > Got it. Thanks for clarifying.

---

### Official Review · Reviewer_d8SV · 2025-07-03

**Clarity:** 3
**Significance:** 2
**Originality:** 3
**Rating:** 4
**Confidence:** 3

**Summary:**

This work introduces causal path tracing, a method for efficiently identifying causal paths that perform particular computations within a transformer. The work provides a thorough exposition of its methods, goals, and motivations, and evaluates it on a variety of models and on both language and vision tasks. They find that causal path tracing identifies paths that are class-specific in a vision transformer, and that contain fewer instances of self-repair than other paths.

**Questions:**

Why is it explicitly beneficial to omit self-repair mechanisms?
When can this be detrimental?

**Ethical Concerns:**

["NO or VERY MINOR ethics concerns only"]

**Final Justification:**

Thank you for the thoughtful and effortful reply to my review. The additional analyses help address my concerns. Please include these additional results in the camera ready, including the omitted recall results from the ground truth IOI analysis. While I understand running a quick version of the analysis for a rebuttal period, including the more complete version will only help the paper. I have raised my score to a 4 in light of these extended analyses.

**Limitations:**

Yes

**Quality:**

2

**Strengths And Weaknesses:**

The work provides extensive descriptions of its motivations and method, and provides a helpful theoretical analysis regarding its runtime. However, some more standard notation might be helpful, especially in the set of equations provided in (1). This work also presents empirical results from both language and vision, which is fairly rare amongst mechanistic interpretability papers.

The work can be improved in a few ways. First, it would be helpful to include greater exposition on the self-repair score. How is this computed, and why does it matter? Would a complete functional description of the causal mechanisms implicated in a particular behavior not also include backup mechanisms? Additionally, it would be a boon to the paper to include a causal path tracing results for circuits with known, manually annotated ground truths (such as IOI, Greater-Than, Docstring, Subject-Verb agreement, etc). Reporting the overlap or precision/recall between the discovered paths and manually annotated components would be helpful for contextualizing this work with other existing papers.

Finally, it would be useful to include stronger edge-based baselines, including EAP-IG (Hanna 2024).

With respect to writing, it would be helpful to describe why the intervention strategies described in the paragraph starting on line 119 violate the conditions in Def. 5, rather than just stating that they do.

---

> ### Author Rebuttal · Authors · 2025-07-31
>
> We sincerely thank the reviewer for the valuable and constructive feedback, which helped improve the clarity and rigor of our paper.
>
> ### **[R1] About Self-repair**
>
> **[R1-1] How it is computed**
>
> Building on the notion of self-repair introduced by Rushing and Nanda [1], we define the self-repair score received by each attention head in a given layer as:
>
> $\textup{self-repair} = \Delta logit-\Delta DE_{head}$
>
> Here, $\Delta logit$ measures the change in the logit corresponding to the original decision after ablating the attention head, $\Delta DE_{head}$ measures the *direct effect* of the head, i.e., its original contribution to the decision logit before any downstream self-repair.
>
> Specifically, the final residual stream, i.e., the classifier input, contains the sum of outputs from all heads across all layers. Accordingly, the contribution of each head can be directly approximated by projecting its output onto the classifier weight vector; this yields the direct effect of that head, $DE_{head}$. Furthermore, following the assumption in [1] that an ablated head contributes zero to the decision, we compute $\Delta DE_{head} = -DE_{head}$.
>
> [1] Cody Rushing and Neel Nanda. Explorations of self-repair in language models.
>
> **[R1-2] Clarifying the backup mechanism in our context**
>
> We fully agree that backup mechanisms can play a functional role in maintaining a model's behavior. However, we would like to clarify that, **in our context, the self-repair score reflects how much a target node receives backup from other nodes,** thereby indicating its role as a backup receiver. In other words, there are two roles in a backup mechanism during ablation: a giver node, which supplies the compensatory signal, and a receiver node, which is affected by it. The self-repair score we measure captures the extent to which a node functions as a receiver in this process.
>
> Specifically, for a node to act as a giver, it must be capable of supporting the decision itself; its contribution to the decision must be high.Given that the total logit for the decision is a fixed quantity, greater contribution by givers implies lesser contribution from receivers. In this sense, giver nodes involved in backup mechanisms may be seen as indicative of the causal mechanism itself. Therefore, if the discovered paths contain many receiver nodes, their self-repair scores will be high, indicating a lower contribution to the decision compared to other paths composed primarily of giver nodes; that is, such discovered paths are likely less causal. In contrast, **if receiver nodes mostly lie outside the discovered paths, their self-repair scores within the paths will be lower than those outside, suggesting that the discovered paths contribute more directly to the decision than nodes outside the paths.** Taken together, a low self-repair score indicates that the discovered path contributes more directly to the decision than components outside it, thereby serving as evidence of its causal nature.
>
> ***
>
> ### **[R2] About Further Tracing Results with GT and EAP-IG**
>
> We thank the reviewer for suggesting evaluation using annotated ground truths, which offer valuable context for assessing circuit-based interpretability. While the annotated ground truths and our causal framework may be based on different assumptions, **we find that our method achieves a high precision of 0.727 on the IOI task,** as shown in the below table.
>
> |Method|Precision|
> |-|-|
> |$\text{NT}_1$|0.146|
> |$\text{NT}_{\text{10\%}}$|0.392|
> |$\text{ET}_{\text{all}}$|0.157|
> |$\text{ET-IG}_{\text{all}}$|0.154|
> |$\text{CPT}$|**0.727**|
>
> Here, EAP-IG (Hanna 2024) is referred to as $\text{ET-IG}$; similar to $\text{ET}$, it is defined in two variants: $\text{ET-IG}\_{\text{all}}$ and $\text{ET-IG}\_{\text{cls}}$. Given that the IOI task does not involve class labels, we exclude $\text{ET}\_{\text{cls}}$ and $\text{ET-IG}\_{\text{cls}}$ from this comparison. Furthermore, we note that, due to the limited rebuttal period, we restricted our minimality-based path tracing method to step 3, i.e., subsets of size up to 3. As a result, some causal nodes that belong to larger subsets may not have been discovered and are therefore counted as false negatives. To ensure a reasonable comparison under these constraints, we report only precision, which reflects the correctness of the identified nodes. Also, the evaluation was conducted on 20 randomly selected samples, due to time limitations.
>
> Interestingly, $\text{NT}_{\text{10\%}}$, which showed relatively low performance on the benchmarks in the main paper, exhibited high overlap with the ground-truth circuit in the IOI task, while ET-based methods showed the opposite trend. Despite these shifts, our method maintained high precision, indicating that the task-relevant components identified in the IOI benchmark are closely aligned with our causal criteria.
>
> We additionally report results on the main paper's benchmark (Table 2), including EAP-IG (i.e., $\text{ET-IG}$), in the table below.
>
> |Method|Hit. ($\uparrow$)|Faith. ($\uparrow$)|Spars. ($\downarrow$)|
> |-|-|-|-|
> |$\text{NT}_1$|0.0000|0.0005|0.6571|
> |$\text{NT}_{\text{10\%}}$|0.0000|0.0006|**0.5648**|
> |$\text{ET}_{\text{all}}$|0.2079|0.2354|0.9806|
> |$\text{ET}_{\text{cls}}$|0.4808|0.4734|0.9909|
> |$\text{ET-IG}_{\text{all}}$|0.6677|**0.7496**|0.9938|
> |$\text{ET-IG}_{\text{cls}}$|**0.7351**|0.7490|0.9923|
> |$\textbf{CPT}$|**0.9826**|**0.5466**|**0.8641**|
>
> EAP-IG achieved the highest hit rate among prior methods, and reported the highest faithfulness score. However, the fact that its hit rate not being close to 1 implies that, when retaining only the identified path, the model often fails to reproduce the original decision. In this context, a high faithfulness score merely indicates that the logit for the original decision is partially preserved, but not necessarily dominant, **i.e., other logits may become even larger in EAP-IG, leading to a different prediction.** Therefore, **our method remains the most reliable, as it consistently identifies decision paths that preserve the original output.** Moreover, whereas EAP-IG retains nearly the entire model structure like other edge-level methods, our method provides a more compact and reliable explanation.
>
> ***
>
> ### **[R3] About Intervention**
>
> To illustrate the distinctions between intervention methods, we consider a simplified transformer architecture with a single block and a single attention head; thus, the block input $z_{ib}$ corresponds to the original token embedding $z_{token}$. The block output can then be decomposed into four additive path terms, as described in Equation (2) of the main paper: the residual only path node ($v_1$), the Attention only path node ($v_2$), Attention+MLP path node ($v_3$), and MLP only path node($v_4$).
>
> **[R3-1] For Definition 5.a (Causal Structural Isomorphism)**
>
> Consider two intervention scenarios where either $v_1$ or $v_4$ is replaced.
>
> Under TOKEN RESAMPLING, we substitute the original token embedding $z_{token}$ with a different in-vocabulary embedding $z_{token}'$. This substitution affects both $v_1$ and $v_4$, as they both contain $z_{token}$ (i.e., $z_{ib}$) as their root variable. Crucially, whether $z_{token}$ or $z_{token}'$ appears in $v_1$ or $v_4$ leads to distinct expressions for the block output. By these expressions, the resulting interventional graphs remain distinguishable across scenarios. This one-to-one correspondence between the mathematical form and the graph structure holds not only in this toy example but also more generally, thereby satisfying **Definition 5.a.**
>
> In contrast, DIRECT NOISE adds the same Gaussian noise directly to multiple nodes (e.g., $v_1$, $v_4$), which leads to overlapping outcomes across different intervention scenarios. This is due to the additive nature of the block output, where all path node contributions are combined through addition. As a result, this destroys the one-to-one mapping between interventional graphs and their mathematical representations, thereby violating **Definition 5.a.**
>
> **[R3-2] For Definition 5.b (Causal Edge Validity)**
>
> Consider an intervention scenario where all path nodes are replaced.
>
> Under ZERO MASKING, each component is zeroed out, which causes the model output to collapse to a constant vector (i.e., the block output $z_{ob} = \mathcal{0}$), resulting in a fixed decision $y = b_{cls}$, where $b_{cls}$ denotes the classifier's bias term. In this case, the intervened decision becomes entirely independent of the original input, relying solely on the classifier bias. As a result, there exists at least one decision for which the causal necessity condition is never satisfied, thereby violating **Definition 5.b.**
>
> In contrast, TOKEN RESAMPLING prevents the decision from collapsing to the classifier bias $b_{cls}$ by replacing the original token embedding $z_{token}$ with a different in-vocabulary embedding $z_{token}'$, ensuring that the causal necessity condition can still be satisfied and thus upholding **Definition 5.b.**
>
> **[R3-3] For Definition 5.c (Intervention Controllability)**
>
> Consider an intervention scenario where $v_4$ is replaced.
>
> TOKEN RESAMPLING replaces a representation with another token embedding from the model's learned vocabulary. Because these embeddings originate from the model itself, the resulting representations remain on the model's manifold, ensuring that interventions stay within the model's natural distribution and thus satisfying **Definition 5.c.**
>
> In contrast, NOISE TOKEN replaces $z_{token}$ with $z_{token} + z_{noise}$, where $z_{noise}$ is the noise. As this composite input is passed through the MLP in $v_4$, the noise term is transformed by learned parameters, producing an activation that is not guaranteed to lie on the model's representation manifold. Furthermore, as this noisy signal propagates through subsequent layers, this may lead to out-of-distribution representations, thereby violating **Definition 5.c.**

---

> > ### Comment · Reviewer_d8SV · 2025-08-06
> >
> > Thank you for the thoughtful and effortful reply to my review. The additional analyses help address my concerns. Please include these additional results in the camera ready, including the omitted recall results from the ground truth IOI analysis. While I understand running a quick version of the analysis for a rebuttal period, including the more complete version will only help the paper. I have raised my score to a 4 in light of these extended analyses.

---

### Comment · Area_Chair_Rmcs · 2025-08-04
**Less than 3 days remaining for Author-Reviewer discussion period**

This is a reminder that the author-reviewer discussion period is meant to afford the time for a proper discussion. Since the authors have devoted substantial effort to their response, I encourage all of the reviewers to critically engage with the response, which includes reading the other reviews and having an open discussion with the authors.

Thanks all for their valuable efforts so far, and please continue helping to ensure high-quality reviewing at NeurIPS for this year and future years!

---

### Comment · Area_Chair_Rmcs · 2025-08-05
**Notes on reviewer participation**

Hi reviewers,

First, thank you to those of you who are already participating in discussions! Your engagement is invaluable for the community and helps to ensure that NeurIPS is able to continue as a high-quality conference as it grows larger and larger every year.

I want to highlight some important points shared by the Program Chairs. First, the PCs have noted that many reviewers submitted “Mandatory Acknowledgement” without posting participating in the Author-Reviewer discussion, and we have been instructed that such action is **NOT PERMITTED**. As suggested by the PCs, I am flagging non-participating reviewers with “InsufficientReview”; I will remove the flag once reviewers have shown an appropriate level of engagement.

Here is a brief summary of the PC’s points on this matter:
- It is not OK for reviewers to leave discussion till the last moment.
- If the authors have resolved your questions, do tell them so.
- If the authors have *not* resolved you questions, do tell them so too.
- The “Mandatory Acknowledgement” button is to be submitted only after the reviewer has read the author rebuttal and engaged in discussions - reviewers **MUST** talk to the authors and are encouraged to talk to other reviewers.

To facilitate these discussions, the Author-Reviewer discussion period has been extended by 48 hours till August 8, 11:59pm AOE.

Thank you all for your efforts so far, and I look forward to seeing a more engaged discussion in the coming days!

---

### Note · Authors · 2025-08-12

We sincerely thank the AC and all reviewers for their time and constructive effort throughout the review and discussion process.

Our work is the first to trace **full-coverage causal paths for a given decision in a transformer** with polynomial-time complexity on average, supported by both theoretical guarantees and extensive empirical validations. We also provide interpretive insights across both language and vision models, showing that nodes along causal paths receive less self-repair, indicating greater direct contribution to the decision, and that certain internal causal paths exhibit class-specific responses.

Put simply, to the best of our knowledge, no prior work has achieved both **reliability**, by considering all possible decision paths, and **feasibility**, by ensuring computational tractability in practice. **Our method offers both, which we believe is the basis of its novelty.**

Following the discussion, **all reviewers acknowledged the value of our contribution and raised their ratings toward acceptance.** This shift was enabled by clarifying key concepts, adding further empirical results, and detailing practical applications to downstream tasks. These will be incorporated into the final version, enhancing the quality of the paper.

In closing, as transformer-based models continue to advance and become integrated into our daily lives, we believe that the efficient and reliable framework proposed in this work represents an important step toward the safe and trustworthy use of AI. We hope that a positive final decision will make our research a foundation for open directions in future work in this field.

---

### Decision · Program_Chairs · 2025-09-17

**Decision:**

Reject

**Comment:**

**Summary:** The paper proposes a framework for *path tracing* in transformers, using a causal perspective to understand how information flows through the network to determine the final output, i.e., the paper sits within the field of circuit discovery in mechanistic interpretability. The method unfolds blocks of the transformer into nodes in a graph and uses a minimality-based search procedure to find possible causal paths within each block, giving a polynomial-time algorithm.

**Strengths:** Reviewers found the proposed method to be an efficient solution to the problem of circuit discovery, and generally agreed that both the problem and the causal techniques are well-motivated.

**Weaknesses:**
1. *Clarity.* The primary weakness of the work is a lack of clarity. The lack of clarity was specifically highlighted by two reviewers (Rev. 6s2J: "Given the poor level of clarity of the initial writeup, I think another round of review would be needed for fully assessing the validity of the paper", Rev. 3ZGp: "...as I read through the paper, I found myself increasingly confused rather than excited."), while other reviewers lowered their confidence scores to reflect that the paper remains unclear to them. The authors provided extensive clarifications in the rebuttals which were appreciated by the reviewers.
2. **Evaluation.** The second most significant weakness of the work regards how the method is evaluated. Rev. d8SV suggested that the authors compare their circuits to known, manually annotated ground truths, e.g. for the Indirect Object Identification (IOI) task. The authors included this analysis in their rebuttal, showing high precision compared to baselines (but not reporting recall). Other reviewers (Rev. KqMq and Rev. s7J4) thought that the experiments were too limited in scope, only demonstrating the method's use for interpretability in classification without demonstrating how it can assist downstream tasks or for other kinds of models. As the authors point out in their rebuttal, evaluating on downstream tasks and extending beyond classification are important, but it is fair to save these for future work, and the current evaluation is already in line with prior works.

**Conclusion:** Overall, the paper seems both interesting and promising. However, given the weakness described above, there are too many changes needed to properly assess the validity of the paper. Most significantly, I encourage the authors to devote significant efforts to taking the clarifications given during the rebuttal period and integrating them into the paper. I also believe that a more thorough version of the new experiments in the rebuttal (which include both new metrics and a new baseline) will enrich the paper.